# Application of Multigene Panels Testing for Hereditary Cancer Syndromes

**DOI:** 10.3390/biology11101461

**Published:** 2022-10-05

**Authors:** Airat Bilyalov, Sergey Nikolaev, Leila Shigapova, Igor Khatkov, Anastasia Danishevich, Ludmila Zhukova, Sergei Smolin, Marina Titova, Tatyana Lisica, Natalia Bodunova, Elena Shagimardanova, Oleg Gusev

**Affiliations:** 1Institute of Fundamental Medicine and Biology, Kazan Federal University, 420008 Kazan, Russia; 2The Loginov Moscow Clinical Scientific Center, 111123 Moscow, Russia; 3Centre for Strategic Planning of FMBA of Russia, 119121 Moscow, Russia; 4Graduate School of Medicine, Juntendo University, Tokyo 113-8421, Japan; 5Endocrinology Research Centre, 117036 Moscow, Russia

**Keywords:** hereditary cancer, multigene panels, NGS

## Abstract

**Simple Summary:**

Hereditary cancer predisposition syndromes (HCPS) are caused by mutations of a single gene and constitute 5–10% of all cancer cases. HCPS are characterized by early manifestation and the presence of cancer cases in family history. Early identification of genetic predisposition to cancer is crucial for both the patients and their relatives at risk, as it can guide the choice of a treatment strategy for the patients and design personalized surveillance and prevention strategies for family members at risk. The wide use of next-generation sequencing (NGS)-based approaches has facilitated ubiquitous integration of targeted sequencing into clinical practice. Multigene panel testing of cancer predisposition genes is now considered to be a major approach for identification of clinically significant variants in individuals of high risk. This study aims to evaluate the landscape of HCPS-associated genetic variants in Russian individuals with personal and/or family history of cancer using NGS-based multigene panel testing.

**Abstract:**

Background: Approximately 5–10% of all cancers are associated with hereditary cancer predisposition syndromes (HCPS). Early identification of HCPS is facilitated by widespread use of next-generation sequencing (NGS) and brings significant benefits to both the patient and their relatives. This study aims to evaluate the landscape of genetic variants in patients with personal and/or family history of cancer using NGS-based multigene panel testing. Materials and Methods: The study cohort included 1117 probands from Russia: 1060 (94.9%) patients with clinical signs of HCPS and 57 (5.1%) healthy individuals with family history of cancer. NGS analysis of 76 HCPS genes was performed using a custom Roche NimbleGen enrichment panel. Results: Pathogenic/likely pathogenic variants were identified in 378 of 1117 individuals (33.8%). The predominant number (59.8%) of genetic variants was identified in *BRCA1/BRCA2* genes. *CHEK2* was the second most commonly altered gene with a total of 28 (7.4%) variants, and 124 (32.8%) genetic variants were found in other 35 cancer-associated genes with variable penetrance. Conclusions: Multigene panel testing allows for a differential diagnosis and identification of high-risk group for oncological diseases. Our results demonstrate that inclusion of non-coding gene regions into HCPS gene panels is highly important for the identification of rare spliceogenic variants with high penetrance.

## 1. Introduction

Malignant neoplasms are one of the leading causes of morbidity and mortality worldwide. According to the experts, the number of cancer cases and the burden on the healthcare system will continue to grow over the next two decades [1]. Approximately 5–10% of cancers cases are associated with hereditary cancer predisposition syndromes (HCPS) [2]. HCPS is caused by mutations of a single gene and characterized by an early manifestation and the presence of cancer cases in family history [2]. Identification of an HCPS brings significant benefits to both the patient and the relatives at risk [3,4,5,6]. As a consequence, the National Comprehensive Cancer Network (NCCN), the American Society of Clinical Oncology (ASCO), the National Institute for Health and Care Excellence (NICE) and other medical professional organizations have developed clinical recommendations for molecular testing and genetic counseling of patients with breast, colon, thyroid and other hereditary cancers [7,8]. In addition, treatment of HCPS cancer patients can be personalized based on their genetic status. For example, identification of clinically significant *BRCA1/BRCA2* variants may influence surgical decisions and affect systemic treatment options. In the long term, the results of genetic testing should underlie the choice of strategies for clinical observation. It will lead to improved prevention of recurrence and secondary tumors. Nowadays, multigene hereditary cancer panel testing is recommended for individuals who have tested negative for a single HCPS and whose personal and/or family history indicates an inherited susceptibility. The composition of gene panels may vary depending on the type of cancer [9]. Due to the fact that there are still cases of an unestablished diagnosis in individuals with clinical signs of hereditary cancer, the evaluation of non-coding deep intronic regions of the HCPS genes is of increasing interest [10,11,12].

The identification of HCPS-associated genetic variants is highly relevant due to the low number of Russian population-based epidemiological studies. An improved understanding of different mutations’ frequencies in cancer patients could also benefit the advancement of personalized medicine. In our study, we have performed genetic testing using a multigene hereditary cancer panel for Russian patients with clinical signs of HCPS and/or family history of cancer.

## 2. Materials and Methods

### 2.1. Study Group

Study cohort included 1117 probands from Russia: 1060 (94.9%) cancer patients with clinical signs of HCPS and 57 (5.1%) healthy individuals with cancer cases in their family. All patients were consulted by a geneticist and enrolled for molecular genetic testing based on the inclusion criteria (Table 1) that have been derived from National Comprehensive Cancer Network (NCCN) and American Society of Clinical Oncology (ASCO) guidelines [8,9]. All patients provided information about their personal and family history of cancer and signed an informed consent form. The form included information about molecular genetic testing and permission to use their depersonalized data for research and scientific publications. Peripheral blood samples were collected in two EDTA tubes (5 mL each) from all participants. We have partially included data from our previous pilot project to the current analysis [13].

### 2.2. DNA Isolation

Genomic DNA was isolated from whole blood using the DNeasy Blood & Tissue Kit (QIAGEN) according to manufacturer’s protocol and quantified using a Qubit 3.0 fluorometer.

### 2.3. Library Preparation and Sequencing

An amount of 100 ng of isolated DNA was used for sequencing libraries generation by means of KAPA HyperPlus kit (Roche, Basel, Switzerland) using either enzymatic or ultrasonic fragmentation according to manufacturer’s instructions. The size of library fragments was evaluated using Agilent 2100 Bioanalyzer (Agilent technologies, Santa Clara, CA, USA). Quantitative analysis of the final libraries was performed using the Qubit 3 fluorometer (Thermo Fisher Scientific, Waltham, MA, USA) to ensure the equimolar pooling of all sample libraries for future sequencing. Targeted gene enrichment was performed by NimbleGen SeqCap EZ Choice kit (Roche). The gene panel consisted of coding regions and flanking sequences of genes associated with HCPS, based on guidelines from NCCN and ASCO recommendations, as well as genes reported in systematic reviews that assessed their link to cancer [8,9]. All genes included in the panel are listed in Appendix A.

Next-generation sequencing (NGS) was performed on the Illumina MiSeq platform using the MiSeq Reagent Kit v2 reagent kit (500 cycles) (Illumina, San Diego, CA, USA). Quality control for cluster generation, sequencing and alignment was provided by addition of control library PhiX Control v3 (Illumina, San Diego, CA, USA).

### 2.4. Variant Classification and Bioinformatics Analysis

The alignment of the paired end fastq files to a reference sequence (hg38) was performed with the BWA-MEM2 algorithm [6]. Duplicates were marked with Picard MarkDuplicates [14]. The base quality score recalibration and variant calling were then performed using GATK BQSR and GATK HaplotypeCaller, respectively [15]. The interval padding was set to 150 base pairs.

Annotation and interpretation of all identified variants were carried out using internal pipeline and information from various databases (Appendix A). The clinical significance of all identified variants was determined using interpretation standards and guidelines of the American College of Medical Genetics and Genomics and the Association of Molecular Pathology [16]. The effect of single nucleotide variants and indels on splicing was investigated in silico using SpliceAI [17]. All information about variants used for their classification was stored, systematized and updated in the internal database.

## 3. Results

In total, we found that 33.8% (378/1117) of the overall group enrolled had pathogenic (P) or likely pathogenic (LP) genetic variants (Table 2). The most common type of genetic variants was frameshift with 208 (55.0%) cases, followed by nonsense and missense variants, comprising 19.0% and 14.6%, respectively. Splicing genetic variants were found in 38 cases and accounted for 10.0% of all identified alterations. The predominant number (226 (59.8%)) of mutations was detected in the *BRCA1/BRCA2* genes primarily associated with breast and ovarian cancer syndromes. Nevertheless, 124 (32.8%) of the clinically significant genetic variants were found in 35 other cancer-associated genes of variable penetrance: 13 (3.4%) variants in high-penetrance genes and 111 (29.4%) genetic alterations in the genes with mild-to-low penetrance (Figure 1). Overall, 21.5% (240/1117) of variants were detected in high-penetrance genes. Notably, 54 (14.3%) unique genetic alterations identified in our cohort had not been previously described in the literature, and nine (16.7%) of these variants were located in *BRCA1/BRCA2* genes.

Additionally, *CHEK2* was the second most commonly altered gene with the total of 28 (7.4%) identified variants: 4 likely pathogenic and 24 pathogenic variants. This gene is characterized by relatively low penetrance and increases the risk of several types of malignancies [18].

It has to be noted that 11 (Appendix A) variants were reclassified from P/LP to variant of uncertain significance (VUS) or benign/likely benign due to additional evidence from large scale studies or databases that have been released in the period from 2019–2022. This has resulted in a 5.9% decrease in the number of the P/LP genetic alterations in our study.

The predominant number of patients in the study were in either breast cancer or ovarian cancer groups: 800 (71.6%) and 209 (18.7%), respectively. We did not describe in detail other groups due to the small number of individuals (see Table 2).

### 3.1. Breast Cancer

In the breast cancer (BC) group, at least one clinically significant genetic variant was identified in 267 out of 800 (33.4%) samples. The largest number of mutations was found in the *BRCA1* gene—41.6% (111/267) of the total number of variants. A p.Gln1756ProfsTer74 variant accounted for 50.5% (56/111) of all *BRCA1* variants. In total, 18.4% (49/267) of P/LP variants affected the *BRCA2* gene. Out of interesting findings, we can note a c.-39-1_-39del variant located outside the 5′UTR region of *BRCA2*. Genetic variants in other cancer-associated genes accounted for 34.1% (91/267) of all clinically significant findings in the BC group. The distribution of genetic variants is shown in Figure 2A.

### 3.2. Ovarian Cancer

In the ovarian cancer (OC) group, 85 clinically significant genetic variants were identified in 40.7% (85/209) of samples. The highest number of variants was found in the following genes: *BRCA1* (52.9%; 45/85) with the most common variant p.Gln1756ProfsTer74 in 26.7% (12/45), *BRCA2* (22.4%; 19/85) and other cancer associated genes constituting 24.7% (21/85) of variants. The distribution of genetic variants in the OC group is shown in Figure 2B.

### 3.3. Healthy Individuals

The group of healthy patients with family history of cancer encompassed 57 individuals. P/LP variants were identified in eight samples; thus, the detection rate was 14.0%. Among the list of genes with clinically significant variants were both high and mild-to-low penetrance genes.

## 4. Discussion

Our study shows a clear benefit of applying a multigene panel for hereditary cancer testing for discovery of new, geographically specific mutations, as well as for identification of cases of non-canonical pathogenic mutations.

Our findings resemble the results of previously published studies. In the article by Tsaousis G. et al., 746 samples were analyzed using the gene panel that included 33 genes [19]. The gene panel in the study also included genes associated with high, intermediate and low cancer risks. The results are the following distribution of cancer-associated mutations in genes: *BRCA1* (31.4%), *BRCA2* (12.2%), *CHEK2* (10.5%) and other HCPS genes (45.9%), which is similar to our data. The deviations in distributions of genetic variants can be explained by the study cohorts, which included patients from different ethnic groups in both studies. The article by Tsaousis G. et al., described the cohort, which was comprised of patients from Greece, Romania and Turkey, whereas ours involved patients with Russian, Tatar and other ethnic backgrounds [20].

In some cases, determining the penetrance of certain genes is difficult due to conflicting interpretations in different studies. For example, mutations in the *PALB2* gene result in an increase of lifetime BC risk up to 41–60% [20]. However, in some articles this gene is reported to have moderate penetrance [21]. Girard E. et al., have reported the association between the type of genetic alterations in *FANCI, FANCL, ATM, ERCC2* and other genes and elevated risk of developing BC: null variants were shown to be associated with higher level of penetrance in comparison to missense ones [22]. Given the ambiguity associated with penetrance interpretation, we referred *PALB2, ATM* and *POLD1*, as well as other genes, to the mild-to-low penetrance group.

The results of our study have revealed several findings of interest, such as a c.-39-1_-39del variant located outside the 5′UTR region of *BRCA2* gene, which is rarely included in the commonly used targeted gene panels. This variant has previously been described in individuals affected by BC [23,24,25,26]. According to in silico algorithms developed to predict the effect of genetic variants on RNA splicing, the c.-39-1_-39del variant is expected to disrupt an acceptor site in intron 1. However, no functional studies have been conducted to date to prove this prediction. Based on these findings and the fact that we were unable to conduct segregation analysis, we classified this variant as LP.

Conventionally used gene panels mainly focus on coding and flanking regions (±1–2 nucleotides) of the genes, thus missing an important number of deep-intronic variants that result in aberrant pre-mRNA splicing [27]. Another reason for the absence of non-exonic regions in the gene panels is a lack of studies focusing on the effects of variants disrupting consensus “cis” sequences; thus, these types of genetic alterations remain underrepresented in large databases that are primarily used for genetic variant interpretation.

The third most frequently altered gene in our cohort was *CHEK2*, which has been previously linked to increased risk of developing most the common gender-specific cancers: breast and prostate [28]. This gene is characterized by incomplete penetrance and differential risk level based on the type of the genetic variant. Boonen R.A.C.M. et al., have reported that truncating *CHEK2* variants (c.1100del, c.444+1G>A) is associated with a two-to-threefold increase in the risk of breast cancer [29]. The clinical reclassification of more prevalent *CHEK2* missense variants from VUS to either benign or pathogenic is hindered by the lack of evidence and conflicting interpretations results of their functional and clinical significance [29]. The main solution to this issue has already been implemented for the *BRCA1* gene, where the spectrum of amino acid substitutions has been functionally assessed, in order to evaluate the effects of these alterations on the gene function [30].

Our results confirm and extend findings of previous studies regarding founder genetic variants in the Russian population [14,31]. Most of the studies focusing on identifying founder mutations are limited to several high penetrance genes (*BRCA1/2* and *CHEK2*), whereas ours extends this list with other genes that fall under both categories: high and moderate-to-low penetrance. *XRCC2* is one of the exemplifications. Germline variants in this gene have been previously described in the context of increased colorectal and BC risks [32]. Based on these findings, i.e., the relatively high prevalence (1.3% out of total number of identified variants) of the c.96del genetic variant and its previous description as a founder mutation in the Polish population [33], we suggest it to be a potential candidate for a founder mutation in the Russian population; however, additional evidence is required.

Another interesting finding in our cohort is the high incidence of monoallelic variants in other Fanconi anemia (FA) genes: *FANCA, FANCC, FANCD2, RAD51C, FANCG* and *FANCI*. The association between heterozygous genetic alterations in these genes and cancer risk has been intensively investigated; however, the results of available studies remain contradictory. Several groups of scientists have reported the absence of correlation between heterozygous genetic variants in FA genes and elevated cancer risk [34], while other studies indicate a significant increase in the risk of developing cancer for certain variants [35]. The consensus on this matter has not yet been reached, mainly due to either small sample sizes or lack of molecular characterization in previous epidemiological studies. We identified the c.1182_1192delTGAGGTGTTTTinsC (p.Glu395TrpfsTer5) variant in the *FANCG* gene in a 35-year-old male patient with a metachronous testicular germ cell tumor and squamous cell tongue cancer. This variant was previously described in two articles with FA, but the meaning in the context of cancer has not been presented yet [36,37]. In our study, this variant could not be classified as causative.

The next interesting finding was an identification of 12 (3.2%) variants in *POLG*. This gene has been previously described as the cause of an autosomal recessive mitochondrial DNA depletion syndrome (OMIM #203700, #613662), recessive mitochondrial recessive ataxia syndrome (OMIM #607459), progressive external ophthalmoplegia autosomal recessive 1 (OMIM #258450) and autosomal dominant 1 (OMIM #157640). Singh B. et al., demonstrated altered genetic and epigenetic regulations of *POLG* in human cancers and suggested a role for *POLG* germline variants in promoting tumorigenesis [38]. In addition, Wu J. et al., proposed candidate prostate cancer predisposition genes in the Asian population, *POLN* and *POLG*, which had not previously been reported in the Western population in this regard [39].

It is evident from our findings that there is a significant proportion of variants that are absent from large databases. In our study, we have identified a notable number of variants that had not previously been described in the literature; consequently, they cannot be univocally assigned to a certain class of pathogenicity. Population-level studies are highly important, as they increase the amount of evidence and bring attention to these kinds of variants, which as a consequence result in a higher chance of correct diagnosis. Notably, our findings suggest a high chance of identifying a clinically significant genetic alteration in moderate-to-low penetrance genes that have been previously suggested to play a role in cancer etiopathogenesis [40]. These are characterized by the absence from the main spectrum of clinical recommendations, thus increasing a chance of not being reported in the case of variants with limited clinical evidence. This suggests another advantage of multigene panels, as they increase the amount of data that can be utilized for the evaluation of risk associated with pathogenic variants in moderate-to-low genes. In this regard, we propose an extension of the general genetic panel to assess genetic variants in the full range of cancer-associated genes. Evaluation of the roles that these genes play in cancer etiopathogenesis is crucial to the development of therapeutic targets for treatment and prevention of different cancer types.

## 5. Conclusions

In this study, we demonstrated the importance of extending the list of regions included into conventional cancer gene panels, especially for genes with high penetrance. Determining the clinical significance of genetic variants located in mild-to-low penetrance genes requires further investigation with the use of larger cohorts. We anticipate that the accumulation of studies that utilize multigene panel testing will increase the amount of available data, shedding light on the involvement of additional genes with variable penetrance in the etiopathogenesis of cancer.

## Figures and Tables

**Figure 1 biology-11-01461-f001:**
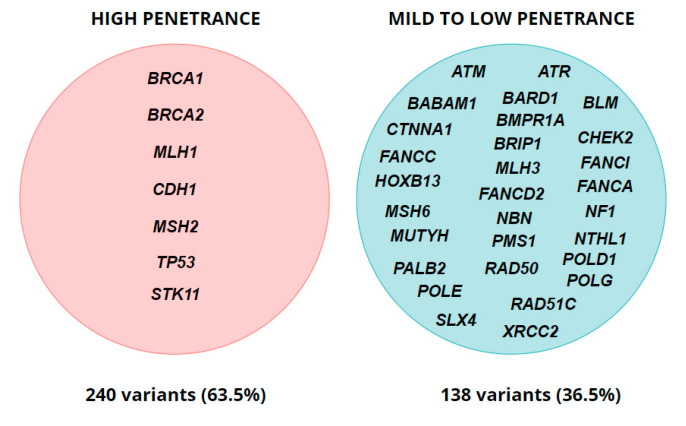
Penetrance of genes with identified P/LP variants (for monoallelic alterations).

**Figure 2 biology-11-01461-f002:**
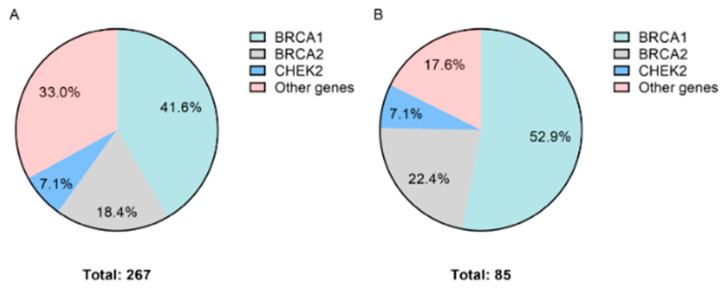
Distribution of genetic variants. (**A**) Breast cancer. (**B**) Ovarian cancer.

**Table 1 biology-11-01461-t001:** Selection criteria for molecular genetic testing.

	**Inclusion Criteria:**	
**№**	**Criteria:**	**Age (years)**
1.	Ovarian cancer	all
2.	Breast cancer:	
	- Female	≤50
	- Triple-negative molecular subtype	≤60
	- Male	all
3.	Colorectal cancer	≤50
4.	Gastric cancer	≤50
5.	Pancreatic cancer	all
6.	Uterine cancer	≤50
7.	Primary multiple tumors (≥2 cancers of one individual):	
	Synchronous	all
	Metachronous	age of first diagnosis ≤ 50
8.	Healthy individuals with ≥1 relatives (parents, children, brothers, sisters) with cancer	all
9.	Healthy individuals with more than 10 colon polyps	all
**Exclusion criteria**		
**№**	**Criteria** **:**	**Age (years)**
1.	Refusal to participate in the study	all
2.	Poor genomic DNA quantity (≤20 ng/µL)	all

**Table 2 biology-11-01461-t002:** Clinical characteristics and frequency of P/LP variants among tested individuals.

Diagnosis	Total Cases	Cases with P/LP Variants (%)	Gene with P/LP Variants	Number of Variants (%)
Breast cancer	800	267 (33.4)	*ATM*	10 (3.7)
			*ATR*	2 (0.7)
			*BARD1*	6 (2.2)
			*BLM*	3 (1.1)
			*BRCA1*	111 (41.6)
			*BRCA2*	49 (18.4)
			*BRIP1*	1 (0.4)
			*CHEK2*	19 (7.1)
			*CTNNA1*	2 (0.7)
			*FANCA*	1 (0.4)
			*FANCC*	3 (1.1)
			*FANCD2*	1 (0.4)
			*FANCI*	11 (4.1)
			*HOXB13*	2 (0.7)
			*MLH1*	1 (0.4)
			*MLH3*	1 (0.4)
			*MSH6*	3 (1.1)
			*MUTYH*	4 (1.5)
			*NBN*	2 (0.7)
			*NF1*	1 (0.4)
			*NTHL1*	2 (0.7)
			*PALB2*	9 (3.4)
			*PMS1*	2 (0.7)
			*POLE*	2 (0.7)
			*POLG*	10 (3.7)
			*RAD50*	1 (0.4)
			*RAD51C*	2 (0.7)
			*SLX4*	1 (0.4)
			*TP53*	2 (0.7)
			*XRCC2*	3 (1.1)
Gastric cancer	13	2 (15.4)	*CHEK2*	1 (50.0)
			*XRCC2*	1 (50.0)
Colon cancer	27	6 (22.2)	*BMPR1A*	1 (16.7)
			*MLH1*	3 (50.0)
			*MSH2*	2 (33.3)
Ovarian cancer	209	85 (40.6)	*ATM*	2 (2.4)
			*BABAM1*	1 (1.2)
			*BLM*	1 (1.2)
			*BRCA1*	45 (52.9)
			*BRCA2*	19 (22.4)
			*BRIP1*	1 (1.2)
			*CDH1*	1 (1.2)
			*CHEK2*	6 (7.1)
			*MSH6*	1 (1.2)
			*MUTYH*	2 (2.4)
			*POLG*	2 (2.4)
			*RAD51C*	3 (3.5)
			*XRCC2*	1 (1.2)
Pancreatic cancer	10	4 (40.0)	*ATM*	1 (25.0)
			*BLM*	2 (50.0)
			*MLH1*	1 (25.0)
Healthy individuals (see Table 1)	57	8 (14.0)	*BLM*	2 (25.0)
			*BRCA1*	1 (12.5)
			*CHEK2*	1 (12.5)
			*MLH3*	1 (12.5)
			*PMS1*	1 (12.5)
			*POLD1*	1 (12.5)
			*STK11*	1 (12.5)
Multiple primary cancers	8	6 (75.0)	*BRCA2*	1 (16.7)
			*CDH1*	1 (16.7)
			*CHEK2*	1 (16.7)
			*FANCG*	1 (16.7)
			*MSH2*	1 (16.7)
			*TP53*	1 (16.7)

P—pathogenic; LP—likely pathogenic.

## Data Availability

The data analyzed in this study is not publicly available. Requests to access the datasets should be directed to the following email: s.nikolaev@mknc.ru.

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
