# Peer review of "Application of Multigene Panels Testing for Hereditary Cancer Syndromes"

_biology, 2022, doi:10.3390/biology11101461_

Round 1
Reviewer 1 Report
Bilyalov et al. present a method to perform multigene panel testing to aid in the diagnosis and identification of individuals with oncogenical disease risk. The authors claim that the inclusion of non-coding gene regions for identification of Hereditary Cancer Predisposition Syndromes (HCPS) is important for rare spliceogenic variants with high penetrance. However, there are several points that need to be addressed to provide more support to their claims.
Major Points:
1) How might the ethnicity and geography of the population impact these studies? Do the authors expect the distribution of mutations to be similar in populations of different ethnicities and geographies? Why or why not?
2) How were the selection criteria chosen for molecular genetic testing in Table 1 and classification criteria for pathogenic and likely pathogenic genetic variants? Please describe or provide references.
3) Have the authors performed any additional analysis to check for any correlation between family history/age/stage of cancer and the gene variants observed?
4) Does Ovarian cancer show non-coding mutations as well?
5) Do the BRCA1 and CHEK2 gene variants have different mutations in healthy individuals? Would there be a way of distinguishing healthy individuals predisposed to cancer and diseased individuals based on the type of mutation?
Minor Points:
1) In line 42, it should be “HCPS is caused…”
2) Please maintain consistency with a single period after the citation. For example, lines 45 and 49 have period before and after the citation numbers.
3) Table 1 has incorrect numbering. In exclusion criteria, it should be “all” to maintain small letter consistency throughout the table.
4) In line 74, provide more details about volume of blood collected and method of collection.
5) In line 84, provide citation for how the gene panel was selected.
6) Line 133 has two periods at the end of the sentence.
7) In line 135, expand VUS and in line 155, expand BC.
8) In line 145, it should be: ”The results have…”
Author Response
Major Points:
1) How might the ethnicity and geography of the population impact these studies? Do the authors expect the distribution of mutations to be similar in populations of different ethnicities and geographies? Why or why not?
The major difference would be observed in the case of founder mutations specific to certain populations, for example: BRCA1 c.4153delA [PMID: 20223023], BRCA1 c.5266dup [PMID: 15146557], BRCA1 c.185delAG [PMID: 17333477] and NBN c.657_661del [PMID: 27936167]. However, less major differences in frequencies for different populations exist for other genetic variants presented in our study, based on the information from GnomAD database [PMID: 32461654]. We wouldn’t be able to perform a statistical comparison with larger cohorts due to relatively small sample size of our study, thus lack of statistical power.
2) How were the selection criteria chosen for molecular genetic testing in Table 1 and classification criteria for pathogenic and likely pathogenic genetic variants? Please describe or provide references.
Selection criteria have been chosen based on NCCN and ASCO guidelines. The description and references have now been added to the manuscript.Classification of genetic variants has been performed using ACMG/AMP criteria, which has been stated and referenced in the “Variant Classification and bioinformatics analysis” section of materials and methods.
3) Have the authors performed any additional analysis to check for any correlation between family history/age/stage of cancer and the gene variants observed?
No additional statistical comparisons were performed due to lack of family history and clinical information for most of the participants.
4) Does Ovarian cancer show non-coding mutations as well?
No, we haven’t identified any clinically significant non-coding genetic variants in a group of patients with ovarian cancer.
5) Do the BRCA1 and CHEK2 gene variants have different mutations in healthy individuals? Would there be a way of distinguishing healthy individuals predisposed to cancer and diseased individuals based on the type of mutation?
In the context of our study, we wouldn’t be able assess the difference in the mutation types for BRCA1 and CHEK2 genes between healthy and patient groups, due to former only having one mutation per each gene. However, we have mentioned the variable penetrance of CHEK2 genetic variants, based on their type (missense or LOF) in the discussion. For BRCA1, there is currently no evidence supporting variable penetration based on the pathogenic variant type.
Minor Points:
1) In line 42, it should be “HCPS is caused…”. Done
2) Please maintain consistency with a single period after the citation. For example, lines 45 and 49 have period before and after the citation numbers. Done
3) Table 1 has incorrect numbering. In exclusion criteria, it should be “all” to maintain small letter consistency throughout the table. Done
4) In line 74, provide more details about volume of blood collected and method of collection. - Done
5) In line 84, provide citation for how the gene panel was selected. Done
6) Line 133 has two periods at the end of the sentence. Done
7) In line 135, expand VUS and in line 155, expand BC. Done
8) In line 145, it should be: ”The results have Done
Reviewer 2 Report
Review of the manuscript 'Application of multigene panels testing for hereditary cancer syndromes’.
This review report concerns the manuscript by Bilyalov on multigene panels used in hereditary cancer syndromes.
The information presented in the manuscript is clear and important to the medical genetics community.
Introduction
Line 42, please rewrite the sentence to a readable phrase. And what is meant by the addition ‘other concomitant symptoms’? Please give a reference here.
I miss in the introduction a short paragraph on NGS multigene panel analysis. Why is this not introduced here? It is the heart of the information in this paper. Please, give at least a review or one or two crucial papers on using multigene panels in cancer diagnostics.
Materials and methods
Line 71, the table is not clear to me. Why is there not a numerical order in the criteria? After 6, it seems that some criteria were lost, since it continues with number 9. And, under exclusion criteria, the quantity is not the same as the quality. I admit that less than 20 ng/µl is very low, but the quality may be still be fine. Please change to ‘quantity’ or give reason why you indicated ‘quality’ here.
Line 74, ‘from the whole blood’, remove ‘the’ and add ‘a’ before Qubit. Please, go through the complete text accurately for minor English issues. (An Agilent 2100 Bioanalyzer etc.). Please, do not use contractions (line 114 for example).
Lines 78 and further, why is the information on suppliers not consistent? Illumina is indicated completely, whereas Roche is just Roche. Maybe include information more consistently here.
Results
No comments.
Discussion
The reference in line 185 does not seem to be correct. Please, give the proper reference if Georgios is meant here.
Lines 195 and further. This part is not clear. I have the feeling that the authors compare their data to another study by Georgios et al., but the reference is not given and neither is it clear from the first sentence that a comparison is made. I strongly advice to rewrite this part.
Line 219, please use standard referencing (Boonen et al. reported that…). Minor issue.
Line 232, ‘XRCC2 is one of the exemplifications’. Of what? If this part belongs to the previous paragraph, please, do not start a new paragraph here.
Line 251, please rewrite the sentence in a clear fashion (‘In our study…’).
Author Response
Thank you very much for your review.
Introduction
Line 42, please rewrite the sentence to a readable phrase. And what is meant by the addition ‘other concomitant symptoms’? Please give a reference here. Done
I miss in the introduction a short paragraph on NGS multigene panel analysis. Why is this not introduced here? It is the heart of the information in this paper. Please, give at least a review or one or two crucial papers on using multigene panels in cancer diagnostics. Done
Materials and methods
Line 71, the table is not clear to me. Why is there not a numerical order in the criteria? After 6, it seems that some criteria were lost, since it continues with number 9. And, under exclusion criteria, the quantity is not the same as the quality. I admit that less than 20 ng/µl is very low, but the quality may be still be fine. Please change to ‘quantity’ or give reason why you indicated ‘quality’ here. Done
Line 74, ‘from the whole blood’, remove ‘the’ and add ‘a’ before Qubit. Please, go through the complete text accurately for minor English issues. (An Agilent 2100 Bioanalyzer etc.). Please, do not use contractions (line 114 for example). Done
Lines 78 and further, why is the information on suppliers not consistent? Illumina is indicated completely, whereas Roche is just Roche. Maybe include information more consistently here. Done
Results
No comments.
Discussion
The reference in line 185 does not seem to be correct. Please, give the proper reference if Georgios is meant here. Done
Lines 195 and further. This part is not clear. I have the feeling that the authors compare their data to another study by Georgios et al., but the reference is not given and neither is it clear from the first sentence that a comparison is made. I strongly advice to rewrite this part. – Done
Line 219, please use standard referencing (Boonen et al. reported that…). Minor issue. - Done
Line 232, ‘XRCC2 is one of the exemplifications’. Of what? If this part belongs to the previous paragraph, please, do not start a new paragraph here. – Done
Line 251, please rewrite the sentence in a clear fashion (‘In our study…’) Done
See the file
